# 1,3-Dioxolan-4-Ones as Promising Monomers for Aliphatic Polyesters: Metal-Free, in Bulk Preparation of PLA

**DOI:** 10.3390/polym12102396

**Published:** 2020-10-18

**Authors:** Stefano Gazzotti, Marco Aldo Ortenzi, Hermes Farina, Alessandra Silvani

**Affiliations:** 1Department of Chemistry, University of Milan, Via Golgi 19, 20133 Milan, Italy; marco.ortenzi@unimi.it (M.A.O.); hermes.farina@unimi.it (H.F.); alessandra.silvani@unimi.it (A.S.); 2CRC Materiali Polimerici (LaMPo), Department of Chemistry, University of Milan, Via Golgi 19, 20133 Milan, Italy

**Keywords:** PLA, 1,3-dioxolan-4-one, organocatalysis, Brønsted acid, lactide

## Abstract

The first example of solvent-free, organocatalyzed, polymerization of 1,3-dioxolan-4-ones, used as easily accessible monomers for the synthesis of polylactic acid (PLA), is described here. An optimization of reaction conditions was carried out, with *p*-toluensulfonic acid emerging as the most efficient Brønsted acid catalyst. The reactivity of the monomers in the tested conditions was studied following the monomer conversion by ^1^H NMR and the molecular weight growth by SEC analysis. A double activation polymerization mechanism was proposed, pointing out the key role of the acid catalyst. The formation of acetal bridges was demonstrated, to different extents depending on the nature of the aldehyde or ketone employed for lactic acid protection. The polymer shows complete retention of stereochemistry, as well as good thermal properties and good polydispersity, albeit modest molecular weight.

## 1. Introduction

Following the continuous increase of plastic-made commodity manufacturing, research efforts in recent years have focused on the possibility of replacing traditional oil-based polymers with more eco-friendly materials. Polylactic acid (PLA) is one of the most intriguing polymers in this category, being derived from renewable sources and, at the same time, having good biodegradability [1,2]. PLA can be synthesized either through polycondensation of lactic acid under azeotropic distillation conditions [3] or, more efficiently, through ring-opening polymerization (ROP) of lactide. Both routes usually require appropriate catalytic systems. ROP is usually preferred, allowing for an optimal control over both the stereochemistry and molecular weight, yielding polymeric products with highly controlled properties [4]. Therefore, L-lactide bulk ROP, is currently the most common and industrially exploited protocol for PLA synthesis [5], usually relying on tin(II) 2-ethylhexanoate (Sn(Oct)_2_) catalysis.

Even if lactide ROP is extremely effective, the synthesis of lactide monomers as precursors is highly challenging. Lactides can be prepared either through the dehydrative cyclization of parent α-hydroxy acids [6] or, from an industrial perspective, through thermal cracking of oligomers under metallic catalysis. For the thermal cracking a high thermal stability of the starting material is required, and this approach gives rise to non-negligible epimerization, offering limited opportunities as regards the functionalization and targeted chemical modification of the lactide monomer. On the other hand, the dehydrative cyclization of α-hydroxy acids relies on high dilutions and long reaction times, affording the product only in moderate yields. In order to tune and broaden the properties of PLA-based materials, other cyclization strategies have also been reported aimed at preparation of lactide monomers bearing functional groups, based on the use of variously substituted α-hydroxy acids and α-haloacyl chlorides as precursors. However, these reactions proved to be troublesome, because of competition between the desired formation of the lactide six-membered ring and oligomerization [7].

Given both the hurdles in functionalized lactide monomers synthesis as well as the modest driving force for lactide ROP, brought by the relief of the strain of the six-membered ring [8], alternative heterocyclic monomeric species have been developed. In order to increase the ROP driving force 1,3-dioxolane-2,4-diones, also known as O-Carboxy Anhydrides (OCAs), were studied [9,10]. OCA polymerization is indeed very fast, thanks to the release of CO_2_ during the reaction. In addition, it tolerates well the presence of different side groups, allowing the use of functionalized OCAs and hence the introduction of different structural motifs along the polyester backbone [11]. However, OCA synthesis, relies on the use of phosgene or its equivalents, resulting in high cost and serious toxicity. In addition, OCAs intrinsic instability makes them difficult be stored for a prolonged time.

In order to further explore the idea of taking advantage of the release of a small molecule during polymerization in order to increase the driving force, Cairns and coworkers quite recently reported a promising approach to ROP, exploiting the elimination of formaldehyde or acetone [12]. The strategy relies on the use of 1,3-dioxolan-4-ones (DOXs) as a versatile monomeric system derived from sustainable and inexpensive sources [13], with salen-based aluminum complex as Lewis acid catalyst (Figure 1).

Together with the research on new monomers, intensive efforts have been recently devoted to the study of more ‘green’ alternatives for the synthesis of PLA, through the replacement of toxic and polluting heavy metal catalysts with organocatalysts [14,15]. Up to now, various catalytic systems have been developed for the metal-free polymerization of both lactides and OCAs. To this regard, thioureas [16], benzoic acid [17], 1,8-diazabicyclo[5.4.0]undec-7-ene and 1,5,7-triazabicyclo[4.4.0]dec-5-ene [18] represent some of the most effective organocatalysts (Figure 1). However, their application in bulk remains a major challenge, which still prevents the transition to industrial scale [19]. Bulk conditions for lactide ROP with basic organocatalysts have been widely investigated displaying however serious drawbacks, related to their low thermal stability and racemization of L-lactide [20]. On the other hand, acidic organocatalysts have been scarcely explored until now, with only a few recent studies in bulk, exploiting triflic acid (TfOH) [21] and diphenylphosphate (DPP) [22,23] respectively, as Brønsted acid catalysts.

Given our interest in the synthesis of PLA-based materials [24,25,26], we looked at DOXs as promising monomers for a “green” PLA-production [27,28]. A mild DOX-polymerization procedure, exploiting an easily available and low-cost acid organocatalyst, in bulk at 100 °C is described here. The effectiveness of protic acid catalysis for DOX polymerization was recently demonstrated by us, in the preparation of high-performance thermosets, starting from a eugenol-functionalized DOX monomer [27]. A screening of the reaction conditions was carried out, and the resulting material showed high retention of stereochemistry, good dispersity of molecular weights, and good thermal properties, when compared to those reported for standard PLA [29,30]. The optimization of a solvent and metal free protocol, as well as the properties of the obtained PLAs, might allow potential applications in the biomedical field [31,32] to be foreseen. Further studies will be carried out to expand the synthesis to high molecular weight products.

## 2. Materials and Methods 

### 2.1. Materials

All reagents were purchased by Sigma Aldrich (St. Louis, Missouri, USA) and used as received. L-(+)-lactic acid, ≥98%; paraformaldehyde, reagent grade, crystalline; *p*-toluenesulfonic acid monohydrate, ≥98%; toluene, ≥99.7%; acetone ≥99.5%; magnesium sulfate, anhydrous, ≥99.5%; trifluoromethanesulfonic acid, 98%; trifluoroacetic acid, 99%; diphenyl phosphate, 99%; (±)-camphor-10-sulfonic acid, 98%; neopentanol, 99%.

### 2.2. Monomers Synthesis

#### 2.2.1. Synthesis of 1,3-Dioxolan-4-One (MeDOX)

L-lactic acid (5 g, 55.5 mmol), paraformaldehyde (4.17 g, 138.8 mmol) and *p*-toluensulfonic acid (5.36 mg, 0.03 mmol) were dissolved in toluene (150 mL) and refluxed for 6 h using a Dean–Stark apparatus to periodically remove water. The reaction was then cooled and the solvent evaporated. The residue was dissolved in CH_2_Cl_2_ (50 mL) and washed with sodium bicarbonate aqueous solution (3 × 25 mL) and brine (3 × 25 mL). The organic layer was dried with sodium sulfate and the solvent was evaporated to obtain the product as a colorless oil (yield 70%). ^1^H NMR (400 MHz, CDCl_3_) δ 5.55 (s, 1H), 5.42 (s, 1H), 4.30 (q, 1H, J = 6.0 Hz), 1.49 (d, 3H, J = 6.0 Hz). ^1^H NMR spectrum of MeDOX is reported in the Appendix A.

#### 2.2.2. Synthesis of 2,2,5-Trimethyl-1,3-Dioxolan-4-One (Me_3_DOX)

L-lactic acid (5 g, 55.5 mmol) and *p*-toluensulfonic acid (212 mg, 1.23 mmol) were dissolved in a 1:1 acetone:toluene mixture (300 mL) and refluxed for 6 h with a Dean–Stark apparatus to periodically remove water. The reaction was then cooled and solvent evaporated. The residue was dissolved into CH_2_Cl_2_ (50 mL) and washed with sodium bicarbonate aqueous solution (3 × 25 mL) and brine (3 × 25 mL). The organic layer was dried with sodium sulfate and the solvent was evaporated. The product was dried over CaH_2_ overnight yielding a colorless oil (yield 58%). ^1^H NMR (400 MHz, CDCl_3_) δ 4.47 (q, 1H, J = 6.8 Hz), 1.60 (s, 3H), 1.53 (s, 3H), 1.47 (d, 3H, J = 6.8 Hz). ^1^H NMR spectrum of Me_3_DOX is reported in the Appendix A.

### 2.3. Optimized Polymerization Procedure

Monomer, acid catalyst and initiator were mixed under nitrogen atmosphere, without solvent. MgSO_4_ was added, to remove any possible residual moisture from the system. The polymerization reaction was conducted at 100 °C, for 6 h. The reaction mixture was dissolved into CH_2_Cl_2_ (20 mL) and washed with sodium bicarbonate aqueous solution (3 × 15 mL) and brine (3 × 15 mL). The organic layer was dried over sodium sulfate and the solvent evaporated, to give a yellowish solid. This was then dissolved in a minimum amount of CH_2_Cl_2_ and precipitated in MeOH, to obtain the product as a white solid. ^1^H NMR (400 MHz, CDCl_3_): δ 5.15 (q, 1H, J = 4.0 Hz), 1.57 (d, 3H, J = 4.0 Hz).

### 2.4. Characterization

#### 2.4.1. Size Exclusion Chromatography (SEC)

Molecular weights and dispersities of the polymers were evaluated using a SEC system having a Waters 1515 Isocratic HPLC pump and four Waters Styragel columns’ set (HR3-HR4-HR5-HR2) with a UV detector Waters 2487 Dual λ Absorbance Detector set at 230 nm, using a flow rate of 1 mL min^−1^ and 60 μL as injection volume. Samples were prepared dissolving 50 mg of polymer in 1 mL of anhydrous CH_2_Cl_2_ and filtering the solution on 0.45 μm filters.

Given the relatively high loading, a check was performed using lower concentration of polymer (5 mg mL^−1^), in order to ensure no column overloading. However, higher loadings were preferred as the UV signal of PLA is relatively weak.

Molecular weight data are expressed in polystyrene (PS) equivalents. The calibration was built using monodispersed PS standards having the following number average molecular weight.

(Mn) and molecular weight distribution (Ð): Mn = 1600,000 Da (Ð ≤ 1.13), Mn = 1150,000 Da (Ð ≤ 1.09), Mn = 900,000 Da (Ð ≤ 1.06), Mn = 400,000 Da (Ð ≤ 1.06), Mn = 200,000 Da (Ð ≤ 1.05), Mn = 90,000 Da (Ð ≤ 1.04), Mn = 50,400 Da (Ð = 1.03), Mn = 30,000 Da (Ð = 1.06), Mn = 17,800 Da (Ð = 1.03), Mn = 9730 Da (Ð = 1.03), Mn = 5460 Da (Ð = 1.03), Mn = 2032 Da (Ð = 1.06), Mn = 1241 Da (Ð = 1.07), Mn = 906 Da (Ð = 1.12), Mn = 478 Da (Ð = 1.22); Ethyl benzene (molecular weight = 106 g mol^−1^). For all analyses, 1,2-dichlorobenzene was used as the internal reference.

#### 2.4.2. ^1^H NMR Analyses

^1^H NMR spectra were registered with a Bruker Ultrashield 400 MHz (Bruker, Billerica, MA, USA). The chemical shifts are reported in ppm and referred to TMS as internal standard. All samples were prepared by dissolving 6–8 mg of polymer into 1 mL of CDCl_3_.

#### 2.4.3. DSC Analysis

DSC analyses were conducted using a Mettler Toledo DSC1 (Mettler Toledo, Columbus, OH, USA), on samples weighing from 5 to 10 mg. Melting and crystallization temperatures were measured using the following temperature cycles:Heating from 25 °C to 190 °C at 10 °C/min;5 min isotherm at 190 °C;Cooling from 190 °C to 25 °C at 10 °C/min;2 min isotherm at 25 °C;Heating from 25 °C to 190 °C at 10 °C/min.

The first two cycles were run to eliminate residual internal stresses derived from the synthesis and workup procedures. Glass transition temperature (T_g_), cold crystallization temperature (T_cc_), and melting temperature (T_m_) were determined during the second heating scan.

#### 2.4.4. MALDI-ToF

Mass spectra were registered on a Bruker Autoflex MALDI-ToF spectrometer (Bruker, Billerica, Massachusetts, USA). Before the analysis, the sample (1 mg/mL), 2,5-Dihydroxybenzoic acid (DHB) (10 mg/mL) used as matrix, and the salt potassium trifluoroacetate (1 mg/mL) CH_2_Cl_2_ solutions were mixed and drop casted on a stainless steel MALDI plate in a volume of 8 μL.

## 3. Results

### 3.1. Polymerization Reaction of 1,3-Dioxolan-4-One

Our investigation started with the selection of 1,3-dioxolan-4-one (MeDOX) **1** as monomer, known to be able to undergo polymerization through the release of volatile formaldehyde. MeDOX was synthesized as reported in Scheme 1a, with a slightly modified literature protocol [12]. 

The first attempt of the polymerization of **1** was carried out using neopentanol as initiator and TfOH as organocatalyst (1:10:200 initiator:catalyst:monomer molar ratio). The reaction was conducted in bulk conditions, at 100 °C, sampling from the mixture at different times—namely 2, 4.5, 23, 48 h—in order to follow the molecular weight growth. Related SEC chromatograms are reported in Figure 2a.

The reaction took about 48 h to reach full conversion, with a significant growth in terms of molecular weights not earlier than 20 h. The curve relative to the sampling at 48 h shows the presence of high molecular weight products, with very high polydispersity and bimodal profile, accounting for the presence of different macromolecular species. To this regard, the corresponding ^1^H NMR spectrum (Figure 2b) demonstrates the presence of a complex mixture of polymeric species, with some examples proposed in Figure 3. The signal centered at 5.20 ppm is typical for the “**a**” proton in linear isotactic PLA. However, while this signal appears as a sharp quartet in a highly stereoregular polymer, the broad appearance here observed indicates some disturbing elements along the chain, likely due to the occurrence of epimerization phenomena as well as to the contribution of formaldehyde in the polymerization reaction. 

The co-existence of structures like **II**, **III**, and **IV** would broaden the signal centered around 5.20 ppm, accounting also for **c** protons, in a context of increased chain mobility given by the presence of acetal bridges. The complex signal at 1.61 ppm is due to various methyl groups **d** (the signal of the **b** methyl group is visible at 1.51 ppm), and the broad signals centered at 6.99 and 4.56 ppm are attributable to **g**, **e**, **f**, and **j** methylene groups. More regular species such as **V** are likely present, given the sharp appearance of the quartet centered around 4.40 ppm (proton **h**).

As already reported [13], esteracetal bridge statistical introduction along polyester chains can occur and shows effects on the material properties, for example enhancing degradation in aqueous media.

### 3.2. Polymerization Reaction of 2,2,5-Trimethyl-1,3-Dioxolan-4-One

Given the drawbacks related to the use of MeDOX, 2,2,5-trimethyl-1,3-dioxolan-4-one (Me_3_DOX, **2**) was then considered as an alternative; synthesis reported in Scheme 1b. The study of the polymerization started with an overnight reaction, in the same solvent-free conditions described for MeDOX, namely a molar ratio of 1:10:200 initiator:catalyst:monomer. Neopentanol was used again as initiator and TfOH was the catalyst of choice. However, considering that SEC analysis demonstrated that the reaction yielded only oligomeric products, a screening of polymerization conditions was carried out, aimed at obtaining higher molecular weight products.

#### 3.2.1. Catalyst Screening

In the search of suitable organocatalyst for the reaction, a series of protic organic acids were evaluated, testing their different acidic strengths [33,34,35,36]. All reactions were conducted without solvent, under N_2_ atmosphere at 100 °C for 16 h. A 1:10:200 initiator:catalyst:monomer molar ratio was employed as in previous tests. Results are reported in Table 1 and expressed as molecular weight of the final product, evaluated through SEC. Besides the case of TfOH (entry 1), it was possible to describe a correlation between the catalyst’s acidic strength and the efficiency of the polymerization, namely as the acidity of the catalyst increases, the molecular weight grows. When looking at the extremely acidic TfOH, the actual protonation of the monomer is likely followed by the increasing role of the TfO- counterion as chain initiator, which affects the molecular weight of the final product. A similar behavior of TfOH has been already reported when used as catalyst in the ROP of lactide [37].

Reaction catalyzed by DPP (entry 5) resulted in the lowest molecular weight product, possibly because of a limited activation of the monomer by the catalyst. Since *p*-toluensulfonic acid (*p*-TSA) (entry 2) yielded the highest molecular weight polymer, it was selected as the best catalyst for further optimization of the reaction conditions.

As the work proceeds, neopentanol was kept as initiator of choice, given the easy detection of the t-butyl signal through NMR. In particular, it was possible to demonstrate the chain-end incorporation of the alcohol through ^1^H NMR (see spectral expansion reported in Appendix A), as well as through MALDI-ToF.

#### 3.2.2. Catalyst Loading

Two additional reactions were carried out decreasing the loading of *p*-TSA, as reported in Table 2. Alongside the starting 5 m/m% concentration, two tests with lower catalyst loadings were conducted, namely 2 and 0.5 m/m%. While at 2% the outcome of the reaction in terms of molecular weight of the product was only minimally affected, at 0.5% only oligomeric species could be detected, disclosing a limit value for catalyst concentration.

### 3.3. Evaluation of Monomer Reactivity and Molecular Weight Growth

The evolution of the molecular weight was studied employing a 1:10:200 initiator:catalyst:monomer molar ratio with neopentanol as initiator and *p*-TSA as catalyst. Samples were taken from the reaction at 1, 2, 4, 6, and 24 h, following the increase of molecular weight through SEC analysis and the monomer conversion through ^1^H NMR (Figure 4).

SEC chromatograms (reported in Appendix A) show the early formation of oligomeric species, which begin to appear already after two hours. At longer reaction times the peak profile alters its shape and after 24 h it slightly broadens as a result of an increase of polydispersity. In ^1^H NMR spectra, the quartet centered at 4.47 ppm (labeled as **1**) was chosen as the key Me_3_DOX monomer signal, useful to follow its conversion with time.

Starting from the sample at 1 h of reaction time, a quartet centered at 5.19 ppm can be observed. This signal increases its intensity over time and is attributable to the methine proton (labeled as **2**) in the growing PLA chains. Conversion becomes quantitative within 6 h. Spurious signals in the spectra are due to traces of acetone condensation byproduct in the crude product, as well as to unavoidable moisture absorbed during the NMR sample preparation.

Molecular weights and conversions are reported in Table 3. Given the presence of a high number of oligomers in the crude products, peak molecular weight (M_p_) was considered as being more reliable than M_n_ and M_w_. It appears that the monomer fully reacts after 6 h. However, after only 2 h of reaction time the polymeric fraction in the mixture already shows molecular weights close to those of the final product.

The dispersity index was also calculated and is reported in Table 3. From SEC and ^1^H NMR analyses, after 6 h the best combination of conversion, molecular weight and dispersity is reached and can be indicated as the optimal reaction time. Given the relatively high polydispersity observed in crude samples, likely due to the presence of byproducts, the sample at 6 h was purified by means of precipitation in cold methanol, from a solution in CH_2_Cl_2_. A SEC chromatogram of the purified product is reported in Figure 5. It shows a bimodal curve having a shoulder at high retention times (about 2000 s), likely indicating the presence of two different macromolecular species in the product (see later for MALDI-ToF analysis).

A reaction mechanism for Me_3_DOX polymerization catalyzed by *p*-TSA is proposed and compared to that of the 4-(Dimethylamino)pyridine-catalyzed polymerization of LacOCA (Scheme 2A,B). Both reactions proceed with elimination of a small molecule and through a double activation by the organocatalyst, relaying our hypothesis on the well-known ability of sulfonic acids to act through such an activation mode [38,39].

This organocatalyzed polymerization mechanism closely resembles the one that was recently disclosed for the metal-catalyzed ROP of DOX (Scheme 2C) [40].

### 3.4. MALDI-ToF

MALDI-ToF analysis was carried out, in order to fully describe the structure of the product. The spectrum is reported in Figure 6 highlighting the presence of two distributions. The main distribution refers to the PLA chains with bonded neopentanol initiator. Alongside this main distribution, minor polymeric species were also detected. Their structure was described by assuming that, as the polymeric chain grows, ketal bridges can statistically form through the Tishchenko reaction, as already pointed out in the case of MeDOX [40]. This reaction likely occurs mostly at the late reaction stages when the monomer concentration decreases. As ketal bridges form, the chain growth significantly slows down, due to the competition with the acetone insertion equilibrium reaction. This results in the formation of structures as the one reported in Figure 6. The occurrence of this kind of side reaction, proved by MALDI-ToF, can explain the discrepancy between the theoretical and experimental molecular weight of the product.

### 3.5. DSC Analysis

The DSC analysis on the purified product was performed to assess the thermal behavior of the synthesized polymer (DSC thermogram reported in the Appendix A). The product showed a semi-crystalline behavior, with a cold crystallization peak at 100 °C and a melting transition (T_m_) at 125 °C. Glass transition (T_g_) was detected at 36 °C. While the cold crystallization temperature is close to that reported for PLA, T_g_ and T_m_ are quite low, likely because of the moderate molecular weight of the product [41]. As described in the experimental section, the polymerization reaction was carried out at 100 °C and therefore at a temperature close to the cold crystallization transition. Even working below the melting temperature, the monomer is able to react, likely through a solid state-like polymerization process.

### 3.6. ^1^H NMR Analysis

Decoupled ^1^H NMR spectrum of the purified product was recorded to ascertain the stereoregularity of the polymer and is reported in Figure 7. A sharp singlet centered at 5.19 ppm was detected, demonstrating a complete retention of configuration and therefore confirming the negligible presence of epimerization phenomena during the reaction.

## 4. Conclusions

In conclusion, DOX polymerization through an organocatalyzed protocol was described for the first time. MeDOX proved not to be suitable for this kind of reaction, due to the occurrence of extensive formaldehyde insertion to give esteracetal bridges along the chains. On the other hand, Me_3_DOX polymerization turned out to be more efficient, with a limited occurrence of these side reactions. Different organic acid catalysts were screened for Me_3_DOX polymerization, disclosing a dependence of the molecular weights of the product on both the acidic strength and the concentration of the catalyst. A study of the reactivity of the monomer at the optimized conditions was carried out, highlighting its complete conversion after 6 h, to give a PLA product with good dispersity also characterized by high stereoregularity. A plausible reaction mechanism was proposed, and a full structural investigation was carried out by means of MALDI-ToF analysis.

The use of a solvent- and metal-free protocol might already disclose possible future applications of these materials in the biomedical field, where PLAs with similar molecular weights have already been used. Further work is underway, aimed at increasing the final molecular weight of the polymer. Copolymerization reactions with DOX monomers bearing different side groups will also be tested in due time, in order to exploit the green potential of this protocol for more focused applications.

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
