# Peer review of "1,3-Dioxolan-4-Ones as Promising Monomers for Aliphatic Polyesters: Metal-Free, in Bulk Preparation of PLA"

_polymers, 2020, doi:10.3390/polym12102396_

Round 1

Reviewer 1 Report

This paper on the synthesis of PLA using an alternative monomer is very interesting and well written. The synthetic strategy is clear and well presented. Figures 1 and 3 are difficult to read and of low quality and therefore need to be improved. For these reasons I think it deserves to be published in this journal after minor revisions.

Author Response

This paper on the synthesis of PLA using an alternative monomer is very interesting and well written. The synthetic strategy is clear and well presented. Figures 1 and 3 are difficult to read and of low quality and therefore need to be improved. For these reasons I think it deserves to be published in this journal after minor revisions.

Authors wish to thank the reviewer for the positive comments regarding the manuscript.

As requested, Figures 1 and 3 (Figures 2 and 4 in the revised version of the paper) have been reorganized and improved in quality, in order to achieve optimal readability and clarity.

Reviewer 2 Report

The manuscript entitled: “1,3-dioxolan-4-ones as promising monomers for aliphatic polyesters: metal-free, in bulk preparation of PLA,” by Gazzotti et al., is a nice piece of work in the wide-spread field of the development of greener and more sustainable chemistry processes into the Polymer Science.

The results seem to be accurately presented; however, the scientific writing of the work can be significantly improved.

The new monomers prepared, derived from 3-dioxolan-4-ones, named MeDOX and Me3DOX have been characterized by 1H and 13C RMN. The PLA materials derived have been also fully characterized employing standard techniques such as SEC, DSC, 1H- NMR and MALDI-ToF.

Having said this, in my opinion, the results described here does not make a significance and interesting contribution for general chemistry readership since this work does not represent a novel approach, concept or more efficient method on the preparation of PLAs in comparison with those previously described by both metal-based catalysts or organocatalysts.

In this sense, I recommend to the authors to stablish the state of the art with an initially Chart representing those organocatalysts with very good performances, including the experimental conditions.

In addition, softer polymerization conditions should be investigated. For instance, at room temperature. Under bulk conditions, loads of species are active in the ROP of lactides and analogs.

All Tables and Figures through the manuscript should include the experimental conditions in the footnote and captions, respectively.

It should be also quite recommendable to stablish a comparison of the Mn of the polymeric materials produced, using 1H NMR and SEC techniques to assess discrepancies.

The proposed mechanism on Scheme 2 for the metal-free polymerization of Me3DOX should be based on additional experimental evidences such as 1H/13C-NMR and X-ray diffraction analysis of intermediate species, not only in the agreement with the previous bibliography data.

Several references should be updated, such as 1-9, 14-17, 27-28, 30-37.

For all these reasons detailed above, I cannot consider this work suitable for publication in Polymers.

Author Response

The manuscript entitled: “1,3-dioxolan-4-ones as promising monomers for aliphatic polyesters: metal-free, in bulk preparation of PLA,” by Gazzotti et al., is a nice piece of work in the wide-spread field of the development of greener and more sustainable chemistry processes into the Polymer Science.

The results seem to be accurately presented; however, the scientific writing of the work can be significantly improved.

The new monomers prepared, derived from 3-dioxolan-4-ones, named MeDOX and Me3DOX have been characterized by 1H and 13C RMN. The PLA materials derived have been also fully characterized employing standard techniques such as SEC, DSC, 1H- NMR and MALDI-ToF.

Having said this, in my opinion, the results described here does not make a significance and interesting contribution for general chemistry readership since this work does not represent a novel approach, concept or more efficient method on the preparation of PLAs in comparison with those previously described by both metal-based catalysts or organocatalysts.

In this sense, I recommend to the authors to stablish the state of the art with an initially Chart representing those organocatalysts with very good performances, including the experimental conditions.

Authors wish to thank the reviewer for the suggestion. A new Figure (now Figure 1) has been added in the introduction section, with a brief survey of literature state of the art regarding organocatalytic systems for lactides and OCAs polymerization, compared to the metal-catalysed (Al(salen)-derived catalyst) protocol currently employed for Me3DOX polymerization. Literature references have been also added accordingly.

In addition, softer polymerization conditions should be investigated. For instance, at room temperature. Under bulk conditions, loads of species are active in the ROP of lactides and analogs.

Authors agree that milder conditions would be suitable. To this regard, throughout the optimization process, different temperatures were tested and 100 °C was found to be the lowest temperature for the ROP to take place in reasonable time. In particular, as PLA melting point is abudantly above 100 °C, it would have been impossible to run the reaction at room temperature. Even if it would be reasonable to assume that some monomer insertion reactions could occur even at room temperature (even if probably in very long times), the resulting solid products would make the reaction impossible to proceed further. In order to address this problem, the use of a solvent could be a possibility, being able to guarantee a low viscosity of the system throughout the whole polymerization. However, since we aimed at a solvent-free protocol, we are not interested in developing these reaction conditions.

All Tables and Figures through the manuscript should include the experimental conditions in the footnote and captions, respectively.

As the reaction conditions are reported and discussed at the beginning of each paragraph, we prefer to keep them only as such, in order to avoid having overly crowded captions for figures and tables. We checked some recent papers published in Polymers journal and experimental conditions are never reported in footnotes and captions. Anyway, if the referee believes that this is mandatory, we are obviously available for adding them in a future revision of the paper.

It should be also quite recommendable to stablish a comparison of the Mn of the polymeric materials produced, using 1H NMR and SEC techniques to assess discrepancies.

Authors agree with the reviewer on the importance of giving a full description of the molecular weight distribution of the synthesized species. For this reason, we decided to give both an absolute ad a relative evaluation of the molecular weight of the products, exploiting MALDI-ToF and SEC analyses, respectively. In particular, the SEC system used for the analyses works on a calibration curve obtained through the elution of monodisperse Polystyrene standards. For this reason all molecular weights values reported in the manuscript were expressed as PS equivalents.  We decided to rely on the use of MALDI-ToF instead of 1H NMR since it is more accurate and less prone to errors in the discussion of the results, as the evaluation of the chain-end signals through NMR can sometimes lead to wrong interpretations.

The proposed mechanism on Scheme 2 for the metal-free polymerization of Me3DOX should be based on additional experimental evidences such as 1H/13C-NMR and X-ray diffraction analysis of intermediate species, not only in the agreement with the previous bibliography data.

As stated in the manuscript, authors aimed at a mechanism proposal, since literature data extensively discuss the double activation mechanism of sulfonic acids (ref. 38, 39). Authors agree that more experimental data would be necessary in order to fully ascertain the reaction mechanism, but this investigation stands outside the scope of the manuscript. About this issue, we also took into account the works of Cairns and coworkers, who published the polymerization of DOX catalysed by Al salen (cit. number 12 dating 2017) and in a following paper (cit. number 40, two years later) published a study on the mechanism of this polymerization.

Several references should be updated, such as 1-9, 14-17, 27-28, 30-37.

Authors agree with the reviewer and updated the references as requested.

For all these reasons detailed above, I cannot consider this work suitable for publication in Polymers.

Reviewer 3 Report

This manuscript from Gazzotti and coworkers described the metal-free, bulk polymerization of 1,3-dioxolan-4-ones, which can serve as an alternative to the established method for making polylactide. Although this new methodology is quite interesting, the authors please address following comments and suggestions:

1) The authors should double check the language to avoid typo/misspelling.

For example, in the abstract (line 18), the authors mentioned "The polymeric shows complete retention of stereochemistry...". The "polymeric" shall be corrected as "polymer". In the supporting information, "1H NMR spectra" shall be corrected to "1H NMR spectral".

2) At line 77 and 78, the authors mentioned "in the preparation of high-performance thermosets, starting from an eugenol-functionalized DOX monomer." Reference 26 was used for supporting this statement. But this example was actually not mentioned in that paper.

3) In the supporting information, the authors should consider rescale all proton NMR spectrum to at least include the reference chloroform peak.

4) At line 111, the authors mentioned "Monomer, acid catalyst, initiator and MgSO4 were mixed under nitrogen atmosphere...". However, the authors did not explain why MgSO4 was added into the reaction.

5) At line 122, the authors mentioned "with an UV detector Waters 2487 Dual λ Absorbance Detector set at 230 nm...". However, for all SEC plots in the manuscript, the y axis was labeled as Refractive Index. The authors should double check the set up of instruments.

6) At line 130, the authors mentioned "(Mp) and molecular weight distribution (D)...". It's probably better to use those conventional symbols, such as Mn and Ð.

7) In Figure 3, there was a broad peak around 4.6 ppm on the spectrum from the aliquots taken at 6 hours. Could the authors explain its origin and why it disappeared after 24 hours?

8) At line 287, the authors mentioned "As ketal bridges form, the chain growth significantly slows down..." Could the authors explain why this is the case?

Author Response

This manuscript from Gazzotti and coworkers described the metal-free, bulk polymerization of 1,3-dioxolan-4-ones, which can serve as an alternative to the established method for making polylactide. Although this new methodology is quite interesting, the authors please address following comments and suggestions:

1) The authors should double check the language to avoid typo/misspelling.

For example, in the abstract (line 18), the authors mentioned "The polymeric shows complete retention of stereochemistry...". The "polymeric" shall be corrected as "polymer". In the supporting information, "1H NMR spectra" shall be corrected to "1H NMR spectral".

Authors wish to thank the reviewer for the positive evaluation of the work.

Following the reviewer’s indication, the manuscript has been checked and language mistakes addressed.

2) At line 77 and 78, the authors mentioned "in the preparation of high-performance thermosets, starting from an eugenol-functionalized DOX monomer." Reference 26 was used for supporting this statement. But this example was actually not mentioned in that paper.

Authors thank the reviewer for having noticed the discrepancies between citations and references. All the references have now been checked.

3) In the supporting information, the authors should consider rescale all proton NMR spectrum to at least include the reference chloroform peak.

Spectra in the SI file have now been reformatted to include also the solvent peak and improved in quality as well.

4) At line 111, the authors mentioned "Monomer, acid catalyst, initiator and MgSO4 were mixed under nitrogen atmosphere...". However, the authors did not explain why MgSO4 was added into the reaction.

MgSO4 has been introduced in order to keep the reaction conditions as anhydrous as possible. Following the reviewer’s suggestion, we now have added a short explanation in the experimental section of the manuscript.

5) At line 122, the authors mentioned "with an UV detector Waters 2487 Dual λ Absorbance Detector set at 230 nm...". However, for all SEC plots in the manuscript, the y axis was labeled as Refractive Index. The authors should double check the set up of instruments.

Authors thank the reviewer for having noticed the mistake. Now chromatograms have been corrected, both in the manuscript and in the SI file.

6) At line 130, the authors mentioned "(Mp) and molecular weight distribution (D)...". It's probably better to use those conventional symbols, such as Mn and Ð.

Corrections have been made, as requested.

7) In Figure 3, there was a broad peak around 4.6 ppm on the spectrum from the aliquots taken at 6 hours. Could the authors explain its origin and why it disappeared after 24 hours?

An interpretation of the nature of the peak has been added to the manuscript. Authors believe that spurious peaks can be attributed to the unavoidable absorption of moisture during the preparation of NMR samples. In addition, the presence of acetone condensation byproduct must be also taken into account in the interpretation of the crude products’ spectra.

8) At line 287, the authors mentioned "As ketal bridges form, the chain growth significantly slows down..." Could the authors explain why this is the case?

An explanation of this aspect has been introduced in the discussion. In particular, authors believe that, as acetone reacts with the chain-end groups in the growing polymeric chains, its equilibrium reaction becomes competitive with the insertion of further monomer molecules, slowing down the reaction rate. In addition, this phenomenon becomes more significant at late reaction stages, when the monomer concentration decreases.

Round 2

Reviewer 2 Report

In this new version of the manuscript entitled: “1,3-dioxolan-4-ones as promising monomers for aliphatic polyesters: metal-free, in bulk preparation of PLA,” by Gazzotti et al., the authors have made an effort to improve the work initially presented.

However, as I said in my initial evaluation report, in my opinion, these results described here does not make a significance and interesting contribution for the general chemistry readership since this work does not represent a novel approach, concept or more efficient method on the preparation of PLAs in comparison with those previously described by both metal-based catalysts or organocatalysts.

I would consider sending this work to a more specialized journal in the field of sustainable polymers.

Nonetheless, I leave in the Editor’s hands the final decision for the acceptance of the manuscript.

Reviewer 3 Report

Thank you all for replying to previous comments and suggestions. I recommend "Accept in present form".